# Clinical and Imaging Outcomes after Vitamin D Supplementation in Patients with Multiple Sclerosis: A Systematic Review

**DOI:** 10.3390/nu15081945

**Published:** 2023-04-18

**Authors:** Julie Langlois, Damien Denimal

**Affiliations:** 1Faculty of Health Sciences, University of Burgundy, F-21000 Dijon, France; 2Department of Biochemistry, University Hospital of Dijon, F-21000 Dijon, France; 3INSERM U1231, 3 Bd Lattre de Tassigny, F-21000 Dijon, France

**Keywords:** vitamin D, multiple sclerosis, supplementation

## Abstract

The link between vitamin D and multiple sclerosis (MS) has been suggested in epidemiological, genetic, immunological, and clinical studies. The aim of the present systematic review of the literature was to assess the effects of vitamin D supplementation on clinical and imaging outcomes in patients with MS. The outcomes we assessed included relapse events, disability progression, and magnetic resonance imaging (MRI) lesions. The search was conducted using PubMed, ClinicalTrials.gov, and EudraCT databases, and it included records published up until 28 February 2023. The systematic review was reported according to the Preferred Reporting Items for Systematic reviews and Meta-Analyses (PRISMA) 2020 guidelines. Nineteen independent clinical studies (corresponding to 24 records) were included in the systematic review. The risk of bias in randomized controlled trials (RCTs) was analyzed using the Cochrane risk-of-bias tool. Fifteen trials investigated relapse events, and most of them reported no significant effect of vitamin D supplementation. Eight of 13 RCTs found that vitamin D supplementation had no effect on disability [assessed by Expanded Disability Status Scale (EDSS) scores] compared to controls. Interestingly, recent RCTs reported a significant reduction in new MRI lesions in the central nervous system of MS patients during supplementation with vitamin D3.

## 1. Introduction

Multiple sclerosis (MS) affected 2.8 million people worldwide in 2020 and is the most common disabling neurological disease affecting young adults [1]. MS is an immune-mediated demyelinating disease of the central nervous system characterized by focal perivenular infiltrates of immune cells and plaque formation. Diagnostic criteria for MS are a combination of clinical, imaging, and laboratory evidence according to the McDonald criteria from the International Panel on Diagnosis of MS [2]. The typical symptoms of MS include muscle weakness, loss of coordination, tremor, fatigue, pain, and visual disturbance. MRI is the most useful imaging test for the diagnosis of MS and provides evidence for the dissemination in space and time during the follow-up of the disease. The typical focal hyperintense lesions observed in the central nervous system of MS patients are searched by the acquisition of T2-weighted fluid-attenuated inversion recovery (FLAIR) MRI sequences (Figure 1) [3]. In addition, the use of gadolinium (Gd)-based contrast agents and T1-weighted MRI sequences is interesting to detect active inflammatory lesions. 

Both genetic variants and environmental factors are associated with MS risk. The main environmental risk factors include obesity, Epstein–Barr virus infection, smoking, and vitamin D deficiency. Vitamin D originates from both dietary sources (vitamin D2 and vitamin D3) and endogenous synthesis in skin after exposure to ultraviolet-B radiation (vitamin D3). Then, vitamin D is hydroxylated in the liver to form 25-hydroxyvitamin D [25(OH)D], subsequently followed by additional hydroxylation in local tissues and the kidneys to yield the active form 1,25-dihydroxyvitamin D [1,25(OH)_2_D], also called calcitriol.

The association between vitamin D and MS has been suggested from epidemiological, genetic, immunological, and clinical data. First, the relationship between MS prevalence and latitude, a major variable of vitamin D status, is well established. More precisely, the prevalence increases by 4.3 cases/100,000 per degree of latitude [6]. In addition, a recent study conducted in two Swedish cohorts reported that low sun exposure increases MS risk, and nearly 30% of the effect was due to vitamin D deficiency [7]. Second, mendelian randomization studies have revealed that genotypes associated with higher vitamin D levels protect against MS [8]. Third, low blood levels of 25(OH)D, the biomarker of vitamin D status, are associated with an increased risk of MS and clinically isolated syndrome (CIS) [9,10]. In addition, low 25(OH)D levels are an important risk factor for MS activity and progression [11]. Last but not least, the immunomodulatory properties of vitamin D are now well established. Vitamin D receptors are expressed in almost every immune cell, and calcitriol exerts pleiotropic actions on several immune processes involved in the pathogenesis of MS [12].

Whether vitamin D plays a causative role in MS activity and progression remains a matter of debate. Interventional trials of vitamin D supplementation in animals and humans are therefore crucial to assess whether vitamin D supplementation is beneficial for MS patients. An excellent review recently summarized the data obtained in animals after supplementation with vitamin D [13]. In brief, vitamin D prevents the development and decreases the severity of experimental autoimmune encephalomyelitis, a well-recognized murine model of MS [14,15,16,17,18,19,20]. Moreover, vitamin D mitigates inflammatory infiltrates, demyelination, and neuron loss in mice with experimental autoimmune encephalomyelitis [14,19].

In the last decade, several clinical trials were conducted in MS patients to evaluate the effects of vitamin D supplementation. In particular, robust data were provided in the last few years by publications based on the randomized controlled trials (RCTs) SOLAR, CHOLINE, and EVIDIMS [21,22,23]. Here, we aimed to conduct a systematic review of data from interventional trials of vitamin D supplementation in MS patients focusing on three major outcomes: relapses, disability, and MRI lesions. Whether vitamin D supplementation prevents incident MS remains another interesting matter of debate, but one that remains outside of the scope of this review. The recently published results of the VITAL RCT showed that a daily supplementation with 2000 international units (IU) of vitamin D3 for a median of 5.3 years decreased the incidence of autoimmune diseases by 22% [24]. In addition, vitamin D intake during early infancy may reduce the incidence of type 1 diabetes [25].

## 2. Materials and Methods

This systematic review follows the PRISMA 2020 guidelines [26].

### 2.1. Eligibility Criteria and Data Items

Table 1 shows the inclusion and exclusion criteria of studies according to the Population, Intervention, Comparison, Outcome, Study (PICOS) format. The review protocol is registered in the PROSPERO registry (#CRD42023411095).

### 2.2. Information Sources and Search Strategy

The search procedure was conducted using PubMed, ClinicalTrials.gov, and EudraCT databases. All records up to 28 February 2023 were identified. The search strategy is detailed in Appendix A.

### 2.3. Selection and Data Collection Process

We independently screened the titles and abstracts and discarded records and studies that were not applicable. Relevant studies were selected regarding the inclusion and exclusion criteria. We resolved disagreements through discussion and consensus. We aimed to investigate possible causes of heterogeneity by collecting data on intervention procedures (dosing, duration of supplementation, form of vitamin D, and administration route). We independently collected all data in Excel sheets for each predefined outcome. No automation tool was used in the process.

### 2.4. Study Risk of Bias Assessment

We independently judged the risk of bias in the included randomized trials using the Cochrane risk-of-bias tool (RoB 2). Disagreements in judgement were resolved through discussion and consensus. The risk of bias is presented in Appendix A.

## 3. Results

### 3.1. Study Selection

The PRISMA flowchart is shown in Figure 2. Twenty-four records were identified using our systematic approach, corresponding to 19 independent clinical trials. The main characteristics of the selected studies are reported in Table 2. Only one international study was identified and included 11 European countries [21]. The remaining studies were conducted in a single country in Europe [22,23,27,28,29,30], the Middle East [31,32,33,34,35,36,37,38], North America [39,40,41], and Australia [42]. Of the 19 selected studies, there were 9 RCTs against placebo, 4 RCTs using a low dose of vitamin D as a control group [22,34,39,40], 1 RCT with routine care as a control group [36], and 5 uncontrolled trials [28,31,32,35,41]. Two of the 14 RCTs were open-label designs [36,40], and the remaining 12 trials were double-blind.

Table 3 presents the patients’ characteristics and the inclusion and exclusion criteria of the studies included in the present systematic review. The numbers of recruited patients were <50 in eight studies, between 50 and 100 in seven studies, and >100 in four studies. The selected studies were conducted in populations of patients with CIS [27], CIS or RRMS [22,32], undefined presentation of MS [30,33,36,38], and, for the majority of studies, in RRMS patients. In the selected trials, participants were recruited according to the serum 25(OH)D level at baseline: <50 nM in two studies [35,36], <75 nM in two studies [22,33], <85 nM in one study [29], between 50 and 125 nM in one study [39], and >100 nM in one study [37]. There were no specific criteria for 25(OH)D levels in the other studies.

Table 4 shows the intervention protocols regarding doses, the dosing frequency, and the duration of the trial. The duration of vitamin D supplementation was <6 months for two trials [32,35], 6 months for seven trials [27,28,33,36,38,39,42], 11 months for one trial [41], 12 months for six trials [21,29,31,34,37,40], 18 months for one trial [22], and 24 months for two trials [23,40]. Vitamin D was administered by the intra-muscular route in one study [38] and by the oral route in all other studies. Lastly, vitamin D was given as 1,25(OH)_2_D3 in two studies [37,41], as 1α(OH)D3 in one trial [33], and in native forms (i.e., vitamin D3 or vitamin D2) in the remaining studies.

### 3.2. Relapses

Fifteen (17 reports) of the 19 trials selected in the systematic review reported results regarding the impact of vitamin D supplementation on relapse in MS patients (Table 5). The 15 trials assessed 554 vitamin D-treated and 456 control patients (368 with placebo, 21 with routine care alone, and 67 with low doses of vitamin D). The ARR and/or the raw number of relapses during the intervention were reported in all of the 15 trials, and the time to first relapse was reported in 2 studies.

Ten controlled trials (corresponding to 342 vitamin D-treated and 300 control patients who completed follow-up) found no effect of vitamin D supplementation on ARR or the raw number of relapses. In particular, the SOLAR trial, which was the largest RCT with 174 RRMS patients in each arm, reported no significant difference in ARR between the placebo group and the group supplemented with 14,000 UI/day of vitamin D3 for one year [21]. The proportion of patients free of relapses was also similar between the two groups. In addition, the recently published RCT EVIDIMS also found a similar relapse rate in MS patients treated for 18 months with either a nutritional dose of vitamin D3 (i.e., 400 IU/d) or 20,400 IU/day [22].

Three trials (186 vitamin D- vs. 144 placebo-treated MS patients) found beneficial effects of vitamin D on relapses. First, Achiron et al. observed that 6 months of 1α(OH)D3 reduced the number of relapses, resulting in a higher proportion of relapse-free patients [33]. This effect was reported at 4 and 6 months of treatment and deteriorated 2 months after drug discontinuation. In an uncontrolled trial by Laursen et al., there was an independent association between the increase in circulating 25(OH)D levels after supplementation with vitamin D3 and a decrease in ARR [28]. Finally, a post-hoc analysis of the CHOLINE RCT, conducted to avoid potential bias due to early dropout, showed a significant reduction in the ARR after vitamin D3 supplementation [23].

Finally, Stein et al. reported a higher proportion of RRMS patients with relapses in those treated with 7000 IU/day of vitamin D2 compared to those supplemented with only 1000 IU/day (36 vs. 0%), but the RCT included only 23 participants [48]. Wingerchuck et al. observed relapses in 27% of MS patients treated with 1,25(OH)_2_D3, but the trial design did not include a control group [41].

### 3.3. Disability

The EDSS score, ranging from 0 to 10 points, is a well-established method for assessing disability in MS patients and monitoring changes in the degree of disability over the course of time. We identified 15 trials reporting changes in the EDSS score after vitamin D supplementation in MS patients (Table 6). They assessed 469 vitamin D-treated patients and 353 control patients (312 with placebo, 9 with routine care alone, and 32 with low-dose vitamin D).

Five RCTs (corresponding to 132 vitamin D-treated and 133 control patients who completed the follow-up) reported that vitamin D had a beneficial effect on the EDSS score [23,29,36,37,40]. In the CHOLINE trial, the progression of the EDSS score was slower in the MS patients supplemented with vitamin D3 compared to those who received the placebo [23]. In the Finnish Vitamin D Study RCT, the EDSS score slightly decreased in vitamin D3-treated patients but not in the placebo group [29]. Shaygannejad et al. found a lower progression of the EDSS score in MS patients taking 1,25(OH)_2_D3 compared to the placebo [37]. Burton et al. observed no difference in the change in EDSS scores between the two groups, but the proportion of MS patients with a progression of the EDSS score was lower in the vitamin D3 group [40]. In addition, one trial showed that in 32 RRMS patients, the EDSS score significantly decreased after 8 weeks of vitamin D treatment, but without randomization against a control group [35].

In eight RCTs, vitamin D supplementation had no effect on the EDSS score compared to control patients (including 334 vitamin D-treated and 298 control MS patients who completed the follow-up) [21,22,27,30,33,34,38,42]. In the SOLAR trial, the proportion of patients who were free from EDSS progression after one year was similar between the groups supplemented with either 14,000 IU daily of vitamin D3 or the placebo [21]. In the EVIDIMS trial, the change in the EDSS score after 18 months of intervention was the same as that in MS patients treated with either 20,400 or 400 IU/day of vitamin D3 [22]. In addition, Achiron et al. reported that 1,25(OH)_2_D3 did not significantly change the EDSS progression compared to the placebo [33].

Lastly, Stein et al. observed a significantly higher EDSS score in MS patients treated for 6 months with 7000 IU/day of vitamin D2 compared to those treated with 1000 IU/day [42].

### 3.4. MRI Lesions

We identified nine clinical trials reporting results relative to MRI lesions after vitamin D supplementation in MS patients (Table 7). They assessed 278 vitamin D-treated patients and 228 control patients (199 with placebo and 29 with low-dose vitamin D). The numbers of new or enlarging lesions were reported in all of these nine trials, and the change in the total volume of lesions was reported in five studies.

Compared with the placebo, vitamin D3 supplementation decreased the number of Gd-enhancing T1 lesions or new/enlarging T2 lesions in the SOLAR and Finnish Vitamin D Study RCTs after one year [21,29]. Camu et al. also found a significant reduction of new T1 lesions after 2 years of treatment with 100,000 IU of vitamin D3 every 2 weeks compared to the placebo [23]. In the EVIDIMS RCT, the MS patients supplemented with 20,400 IU/day of vitamin D3 developed fewer cumulative new Gd lesions compared to those treated with 400 IU/day, without reaching significance [22]. On the contrary, O’Connell et al. found no effect after 6 months of supplementation with vitamin D3 in CIS patients [27].

In the CHOLINE RCT, the total volume of MRI lesions decreased more in the vitamin D3-supplemented group than in the placebo group [23]. In SOLAR, the change in the total volume of T2 lesions was lower after one year of high-dose vitamin D3 compared to the placebo [21]. However, the EVIDIMS and Finnish Vitamin D Study RCTs found no such improvement with vitamin D3 [22,29].

## 4. Discussion

In the present systematic review, we identified 19 clinical trials reporting data on the influence of vitamin D supplementation on relapses, disability, or MRI lesions. Among the studies reviewed here, it is worth noting that the SOLAR and CHOLINE RCTs included the largest number of MS patients, and that they were controlled against placebos and conducted in a double-blind manner [21,23]. The EVIDIMS RCT is also of particular interest because it compares supplementation with a low or high dose of vitamin D3, and it included patients with either RRMS or CIS, which is relatively original [22]. We have to mention that we excluded Jelinek’s study from our systematic review [49]. Although they included a very large number of MS patients (>2000), the data on vitamin D supplementation, relapses, and disability were obtained through retrospective self-reporting, which conferred a high degree of the risk of bias [49].

After reviewing the 15 selected trials reporting results on relapses, we determined that there is no solid evidence that vitamin D supplementation is effective at preventing relapses in MS patients. Overall, 10 controlled trials including 63% of the supplemented MS patients in the selected studies reported that there was a lack of significant effects on relapses following vitamin D supplementation. Among the well-designed RCTs, only the CHOLINE RCT reported a beneficial effect of vitamin D3 [23]. In this study, only clinically active RRMS patients treated with IFN-β1 and with a 25(OH)D concentration <75 nM at baseline were included. The 25(OH)D levels were around 50 nM at baseline in the vitamin D3 and placebo groups and tripled to reach 157 nM after supplementation [23]. In contrast, the SOLAR RCT, which also included clinically active RRMS patients treated with IFN-β, did not find a beneficial effect of vitamin D3 on relapses. It may be possible that vitamin D3 has beneficial effects on relapses only if MS patients are vitamin D deficient before supplementation. However, 75% of the MS participants in the SOLAR RCT also had 25(OH)D blood levels lower than 75 nM, and the 25(OH)D levels progressed similar in SOLAR to what was seen in CHOLINE [21]. In addition, two other RCTs performed in MS patients with vitamin D deficiency at baseline found no beneficial effects of vitamin D3 on relapses [29,34]. It is worth noting that CHOLINE had the longest duration of supplementation (i.e., 2 years), and therefore a time effect cannot be definitively excluded.

A previous review conducted by Hanaei et al. also concluded that supplementing MS patients with vitamin D has no significant effect on the relapse rate [50]. However, they did not include the results of the large and well-designed CHOLINE, SOLAR, and EVIDIMS RCTs. We found that the conclusions were not altered when we added the analysis of these new RCTs. A meta-analysis from the Cochrane Library published in 2018 found that vitamin D3 had no effect on the ARR, obviously without including the results from the SOLAR, CHOLINE, and EVIDIMS RCTs, which were published in 2019 and 2020 [51]. However, we cannot fully exclude to date that 1α(OH)D3 has a specific beneficial effect. While only one RCT investigated the effect of 1α(OH)D3 on relapses, it reported a higher proportion of MS patients free of relapses with 1α(OH)D3 compared with the placebo [33].

Regarding the disability in MS patients, we found the results too heterogeneous to confirm that vitamin D has a beneficial effect on the EDSS score. Indeed, five RCTs reported beneficial effects, whereas eight RCTs did not. The CHOLINE RCT found a moderately but significantly lower progression of the EDSS score in the vitamin D3-treated group compared to the placebo group. However, as already mentioned above, because CHOLINE was the longest-running trial, a time effect cannot be ruled out. Interestingly, no difference in the proportion of patients free of EDSS progression was observed after the one-year supplementation in SOLAR [21]. Consistently, the change in the EDSS score was not significantly different between the MS patients treated with a high or a low dose of vitamin D3 in EVIDIMS. In addition, the controlled trials conducted with 1α(OH)D3 or 1,25(OH)_2_D3 also yielded evidence of a lack of benefit [33,37]. Hanaei et al. concluded that vitamin D had no effect on the EDSS score in a previous systematic review, but without considering the results of the CHOLINE, SOLAR, and EVIDIMS RCTs [50]. The most recent meta-analysis available in the Cochrane Library also showed that one year of supplementation with vitamin D3 did not improve the EDSS score more than the placebo [51]. As already mentioned, this meta-analysis is from 2018 and therefore does not include the results from the SOLAR, CHOLINE, and EVIDIMS trials published in 2019 and 2020.

Regarding the MRI lesions, the present systematic review demonstrates that supplementation with vitamin D3 has beneficial effects in MS patients on the development of new lesions in the central nervous system. The placebo-controlled, double-blind, randomized SOLAR, Finnish Vitamin D Study, and CHOLINE trials reported a reduction in the appearance of new lesions of IFN-treated RRMS patients after supplementation with vitamin D3 for 12 or 24 months [21,23,29]. In addition, a similar trend was observed in the EVIDIMS RCT in patients with RRMS or CIS treated with 20,400 IU daily of vitamin D3 compared to those supplemented with 400 IU/day [22]. Three RCTs in which supplementation lasted 6 months found no effects associated with vitamin D, suggesting that the beneficial effect on new MRI lesions may occur only with more than 6 months of supplementation [27,38,42]. However, it must be mentioned that, beyond the shorter duration, these three RCTs are also unique because they were conducted only in CIS patients [27], or with vitamin D2 [42], or with vitamin D administered by the intramuscular route [38]. Contrary to our present conclusions, the most recent meta-analysis in the Cochrane Library concluded that vitamin D3 had no effect on new MRI lesions after one year of supplementation [51]. However, their conclusions were based on only two RCTs with 12-month time horizons. Since then, the results from the SOLAR, CHOLINE, and EVIDIMS RCTs were published, yielding significant new data that show a reduction in new MRI lesions in patients receiving vitamin D3 supplementation. On the other hand, we considered that the level of evidence suggesting a beneficial effect of vitamin D on the total volume of lesions remains too weak to properly draw conclusions. Indeed, the apparent discrepancies between the results from the CHOLINE and SOLAR RCTs versus the EVIDIMS and Finnish Vitamin D Study RCTs remain to be further investigated.

## 5. Conclusions

The present systematic review aimed to assess the influence of vitamin D supplementation on relapses, disability, and MRI lesions in MS patients. We concluded that vitamin D3 supplementation has a beneficial effect on new MRI lesions. In contrast, we found that there is no robust evidence to date that vitamin D supplementation is effective for preventing relapses or the progression of disability.

Here, we chose to focus on three major outcomes in MS, but other clinical outcomes are worth investigating relative to the quality of life of MS patients. Interestingly, Xie et al. recently concluded in a systematic review of RCTs that vitamin D has beneficial effects on MS-related depression [52]. In addition, the systematic review by Głąbska et al. concluded that most studies suggested that vitamin D supplementation in MS patients has a beneficial influence on the quality of life [53].

## Figures and Tables

**Figure 1 nutrients-15-01945-f001:**
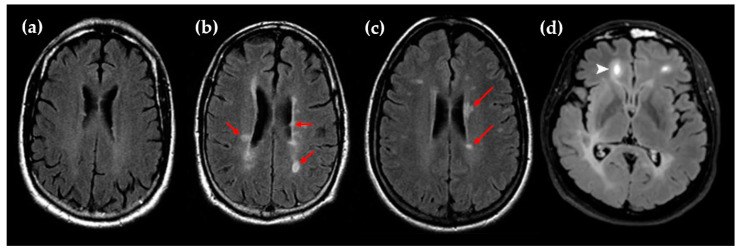
(**a**) Representative MRI scan (T2/FLAIR) of MS patients without brain lesions; (**b**,**c**) Representative MRI scans (T2/FLAIR) of MS patients with numerous brain lesions (red arrows); (**d**) Representative Gd-enhanced lesion in a frontal lobe (white arrow). The images (**a**–**c**) are reproduced from Dastagir et al. with the addition of red arrows (BY-NC-ND/4.0 license) [4]. Image (**d**) is reproduced from Lopaisankrit et al. without changes (CC BY/4.0 license) [5].

**Figure 2 nutrients-15-01945-f002:**
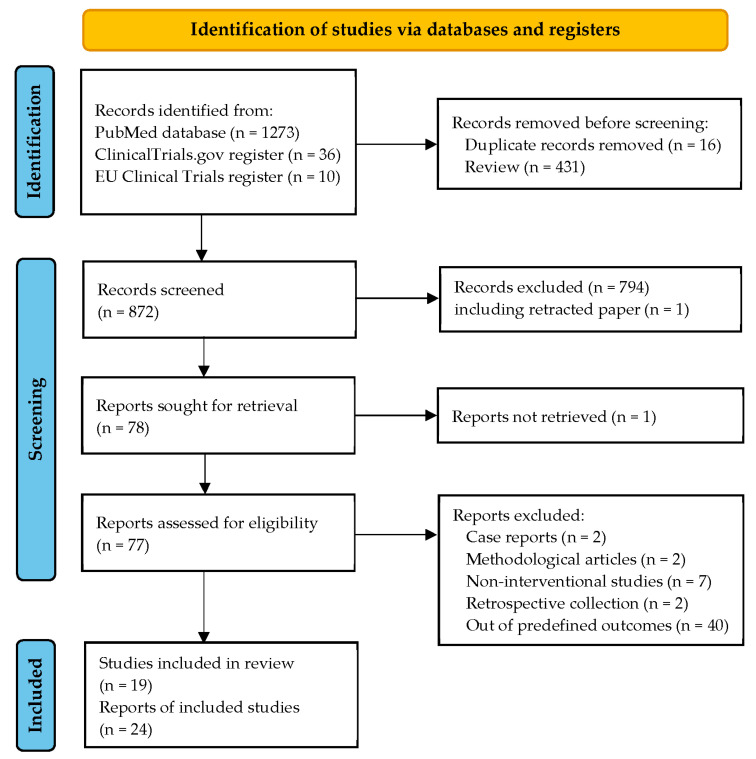
PRISMA 2020 flow diagram for the selection of records and studies.

**Table 1 nutrients-15-01945-t001:** Eligibility criteria for the selection of studies.

PICOS Parameter	Inclusion Criteria	Exclusion Criteria
Population	Adult patients with MS, CIS, or radiologically isolated syndrome (RIS) as defined in the different versions of the McDonald criteria. All subgroups of MS patients [(relapsing remitting multiple sclerosis (RRMS), primary progressive, secondary progressive (SPMS), or progressive relapsing], regardless of gender, disease duration, degree of disability and MRI lesions, and baseline vitamin D levels.	
Intervention	Any form of vitamin D [vitamin D2 or vitamin D3, 25(OH)D3, 1α(OH)D3 or 1,25(OH)_2_D3] with or without associated supplementation in calcium. Any duration, dose, dosing frequency of supplementation. Any administration route.	Supplementation with multiple nutrients (other than calcium). Records were also excluded if data on vitamin D dosing were retrospectively obtained.
Comparison	With a comparison group (placebo, usual care or low-dose use of vitamin D) or without a comparison group (uncontrolled trials).	
Outcomes	Primary outcomes were the changes after vitamin D supplementation in Relapses assessed by the annualized relapse rate (ARR), raw number of relapses, time to first relapse, or proportion of patients free of relapses; Disability assessed by the EDSS score; MRI lesions assessed by the number of new lesions or the total volume of lesions.	Records were also excluded if MS patients retrospectively self-reported data on outcomes.
Study design	Articles/records published in English and in peer-reviewed journals or in recognized databases of trial registration.	Articles not published in English, methodological articles, reviews, meta-analyses, comments, letters to the editor, studies conducted in animals, case reports.

**Table 2 nutrients-15-01945-t002:** Characteristics of selected studies. Sub-studies are grouped with the main study.

Refs.	First Author, Year Trial Registration Number	Country	Study Design	Study Group	Time
[22]	Dörr et al., 2020 NCT01440062	Germany	Randomized, double-blind low-dose controlled	Interferon (IFN)-β1b RRMS/CIS	Not provided
[21,43]	Hupperts et al., 2019 Rolf et al., 2017 NCT01285401	11 European countries	Randomized, double-blind, placebo-controlled	IFN-β1a- treated RRMS	February 2011 to May 2015
[23]	Camu et al., 2019 NCT01198132	France	Randomized, double-blind, placebo-controlled	RRMS	January 2010 to June 2013
[31]	Kotb et al., 2019	Saudi Arabia	Uncontrolled	RRMS	2013–2018
[32]	Darwish et al., 2017 NCT01952483	Lebanon	Uncontrolled	RRMS/CIS	Not provided
[27]	O’Connell et al., 2017 NCT01728922	Ireland	Randomized, double-blind, placebo-controlled	CIS	November 2012 to June 2015
[28]	Laursen et al., 2016	Denmark	Uncontrolled	Natalizumab-treated RRMS	2009–2010
[39]	Sotirchos et al., 2016 NCT01024777	USA	Randomized, double-blind, low-dose-controlled	RRMS	April 2010 to January 2013
[35,44]	Farsani et al., 2015 Naghavi Gargari et al., 2015 IRCT2014011216181N1	Iran	Uncontrolled	RRMS	November 2012 to October 2013
[36]	Etemadifar et al., 2015	Iran	Randomized, open-label, routine care-controlled	Pregnant women with MS	July 2011 to December 2012
[33]	Achiron et al., 2015	Israel	Randomized, double-blind, placebo-controlled	MS	Not provided
[34,45]	Golan et al., 2013a and 2013b NCT01005095	Israel	Randomized, double-blind, low-dose-controlled	RRMS	November 2010 to April 2012
[29,46]	Soilu-Hänninen et al., 2012 Hänninen et al., 2020 NCT01339676	Finland	Randomized, double-blind, placebo-controlled	IFN-treated RRMS	March 2008 to August 2011
[30]	Kampman et al., 2012 NCT00785473	Norway	Randomized, double-blind, placebo-controlled	MS	November 2008 to September 2011
[37]	Shaygannejad et al., 2012 IRCT201104166202N1	Iran	Randomized, double-blind, placebo-controlled	RRMS	October 2007 to October 2008
[42]	Stein et al., 2011 ACTRN12606000359538	Australia	Randomized, double-blind, low-dose-controlled	RRMS	December 2006 to May 2009
[38]	Mosayebi et al., 2011	Iran	Randomized, double-blind, placebo-controlled	IFN-treated MS	October 2009 to April 2011
[40,47]	Kimball et al., 2007 Burton et al., 2010 NCT00644904	Canada	Randomized, open-label, controlled	RRMS	December 2003 to January 2005
[41]	Wingerchuk et al., 2005	USA	Uncontrolled	RRMS	March 1999 to March 2001

**Table 3 nutrients-15-01945-t003:** Characteristics of participants within the studies included in the systematic review.

Refs.	First Author, Year Trial Name	Number of CIS/MS Participants	Age at Baseline	Inclusion and Exclusion Criteria
[22]	Dörr et al., 2020 EVIDIMS trial	53	41 ± 2.1	Inclusion: patients with CIS or RRMS; age 18–65 years (y); EDSS score 0–6; no relapse in the last 30 days; IFN-β1b treatment >3 months. Exclusion: pregnancy; history of sarcoidosis; hepatopathy or renal dysfunction; nephrolithiasis; pseudo-hypoparathyroidism; vitamin D supplementation with more than 500 IU/d in the last 6 months; hypercalcemia or urine calcium/creatinine ratio > 1; concomitant medication with hydrochlorothiazide, digitoxin, digoxin, barbiturates, phenytoin; incompatibility with MRI.
[21,43]	Hupperts et al., 2019 Rolf et al., 2017 SOLAR trial	229	34 ± 8.0	Inclusion: RRMS; age 18–55 y; adequate renal and hepatic function; early-stage MS on brain or spinal MRI; first clinical event in the last 5 years; EDSS score lower than 4.0; active disease with either one relapse or MRI new lesion within the last 18 months; no or low vitamin D supplementation (lower than 1000 IU/d). Exclusion: lactation or pregnancy; other disease than MS that could explain symptoms; relapse in the last 30 days before inclusion; use of corticosteroids within 30 days before inclusion; complete transverse myelitis or bilateral optic neuritis; abnormalities of vitamin D metabolism other than low dietary intake or decreased sun exposure; urinary calcium higher than 1.0 mmol/mmol of creatinine or hypercalcemia (11 mg/dL); hepatic impairment (alanine or aspartate aminotransferase higher than three times the upper limit of normal (ULN); bilirubin higher than 1.5 times ULN if associated with any elevation of alanine aminotransferase or alkaline phosphatase; or alkaline phosphatase higher than 2.5 times ULN); drugs other than corticosteroids that affect vitamin D metabolism; vitamin D supplementation higher than 400 IU/d; conditions with susceptibility to hypercalcemia (e.g., treatment with digitalis or hydrochlorothiazide, arrhythmia or heart disease, nephrolithiasis).
[23]	Camu et al., 2019 CHOLINE trial	129	38 ± 9.3	Inclusion: patients with RRMS; age 18–65 y; serum 25(OH)D < 75 nM; treatment with IFN-β1a for 4 ± 2 months prior to randomization; EDSS 0–5; ≥1 relapse during the previous 2 y; stable disease over the last 30 days. Exclusion: medications affecting vitamin D metabolism other than corticosteroids; previous or ongoing hypercalcemia; estimated glomerular filtration rate (eGFR) lower than 60 mL/min.
[31]	Kotb et al., 2019	35	27 ± 4	Inclusion: patients with RRMS; age ≥18 y; no exacerbations; regular treatment with IFN; no new MRI lesions. Exclusion: glucocorticoid treatment within 4 weeks prior to recruitment; disease-modifying drugs (DMD) other than IFN; vitamin D >1000 UI/d; use of glucocorticoid or relapse in the last 30 days; severe depression; pregnancy; serum creatinine >1.5 mg/dL; hypersensitivity to vitamin D; history of hyperparathyroidism, tuberculosis, nephrolithiasis, or sarcoidosis.
[32]	Darwish et al., 2017	88	34 ± 11	Inclusion: patients with RIS, CIS, RRMS, SPMS; age ≥16 y; 25(OH)D <250 nM. Exclusion: not reported.
[27]	O’Connell et al., 2017	67	37 ± 8.7 (group 10,000 UI/d) 33 ± 4.6 (group 5000 UI/d)	Inclusion: patients with CIS; age 18–55 y; symptom onset in the last 3 months; more than one asymptomatic T2 lesions on brain MRI; no treatment with corticosteroids in the last month; no other DMD. Exclusion: disease other than MS that could explain symptoms; exacerbation in the last six weeks; treatment with any immunomodulating therapy in the last three months; steroids in the last month or any previous treatment with other immunosuppressant or mitoxantrone; no hypercalcemia, renal impairment, intolerance to vitamin D, parathyroid dysfunction, sarcoidosis; pregnancy or lack of contraception; prior or current treatment with thiazide diuretics or vitamin D supplementation (≥1000 IU/d).
[28]	Laursen et al., 2016	134	41 (23–63)	Inclusion: RRMS patients with natalizumab treatment for at least 1 year prior to enrolment. Exclusion: pregnancy; development of SPMS, cancer, anti-natalizumab antibodies.
[39]	Sotirchos et al., 2016	40	41 ± 8.1	Inclusion: RRMS; age 18–55 y; serum 25(OH)D 50–125 nM. Exclusion: vitamin D supplementation >1000 IU; change of immunomodulatory therapy within the past 3 months; systemic glucocorticoid therapy or relapse in the last month; pregnancy; serum creatinine higher than 1.5 mg/dL; vitamin D intolerance; history of hyperparathyroidism, sarcoidosis, tuberculosis, or nephrolithiasis.
[35,44]	Farsani et al., 2015 Naghavi Gargari et al., 2015	32	31 ± 7.1	Inclusion: RRMS; remission period; serum 25(OH)D <50 nM; EDSS score 0–5. Exclusion: treatment with steroid or immunosuppressive drugs.
[36]	Etemadifar et al., 2015	15	27 ± 2.4	Inclusion: pregnant women with MS; age 20–40 y; stable neurological functioning for at least 1 month prior to study entry; EDSS score ≤ 6; serum 25(OH)D level < 50 nM. Exclusion: substantial disorders in psychiatric, hematologic, cardiac, endocrinological, renal, hepatic or metabolic functions; vitamin D3 supplement; any condition predisposing to hypercalcemia; nephrolithiasis; renal insufficiency.
[33]	Achiron et al., 2015	158	41 ± 9.8	Inclusion: fatigue severity scale score ≥ 40; age 18–55 years; EDSS score > 5.5. Exclusion: relapse within 30 days before the study; serum calcium level > 10.5 mg/dL; history of hypersensitivity or intolerance to 1α(OH)D3 or related substances; a life-threatening and/or unstable clinical condition and/or alcohol or drug abuse.
[34,45]	Golan et al., 2013a and 2013b	45	43 ± 12	Inclusion: age ≥ 18 y; patients who continued to suffer from flu-like symptoms after 4 months of treatment with IFN-β; 25(OH)D blood levels lower than 75 nM; EDSS score lower than 7. Exclusion: intestinal malabsorption, cirrhosis, nephrotic syndrome, hyperthyroidism, eGFR less than 40 mL/min, rickets, hypoparathyroidism, hypercalcemia at baseline, known malignancy, granulomatous disorders and lymphomas; treatment with orlistat, anticonvulsants, rifampin, isoniazide, ketoconazole, leucovorin, 5FU, hydrochlorothiazide; arrhythmia; heart disease; nephrolithiasis; pregnancy.
[29,46]	Soilu-Hänninen et al., 2012 Hänninen et al., 2020 Finnish Vitamin D Study	66	39 (range 22–53)	Inclusion: RRMS with IFN-β1b for at least 1 month; age 18–55 y; EDSS score ≤ 5.5; use of a contraceptive method. Exclusion: calcemia > 2.6 mM; serum 25(OH)D > 85 nM; primary hyperparathyroidism; pregnancy or lack of contraception; alcohol or drug abuse; immunomodulatory treatment other than IFN-β1b; intolerance to cholecalciferol or peanuts; treatment with digitalis, vitamin D3 analogs or vitamin D, calcitonin; any condition predisposing to hypercalcemia, sarcoidosis, nephrolithiasis or renal impairment; significant hypertension (higher than 180/110 mmHg); hyper- or hypothyroidism in the last year; a history of nephrolithiasis in the last 5 years; cardiac insufficiency or dysrhythmia; unstable ischemic heart disease; depression.
[30]	Kampman et al., 2012	68	40 ± 8	Inclusion: RRMS; age 18–50 y; EDSS score ≤ 4.5. Exclusion: inability to walk more than 500 m; history of conditions affecting bone; lactation or pregnancy in the last 6 months; use of bone-active drugs other than intravenous methylprednisolone for the treatment of relapse; nephrolithiasis in the last 5 years; menopause; unwillingness to use appropriate contraception.
[37]	Shaygannejad et al. 2012	50	39 ± 8.4	Inclusion: RRMS; age 15–60 y; EDSS score ≤ 6; stable neurological condition for more than 4 weeks prior to study; circulating 25(OH)D level higher than 100 nM. Exclusion: substantial disorders in psychiatric, neurological, cardiac, endocrinological, hematologic, hepatic, renal, or metabolic systems; treatment with digitalis, vitamin D supplementation; any condition predisposing to hypercalcemia, renal insufficiency nephrolithiasis; pregnancy.
[42]	Stein et al., 2011	23	34 [inter-quartile range (IQR) 30–49]	Inclusion: RRMS; age > 18 y; relapse in the last 2 years despite immunomodulatory treatment. Exclusion: primary or SPMS; pregnancy; clinical relapse or use of systemic glucocorticoid within the prior month; EDSS higher than 5; current DMD other than IFN or glatiramer acetate; elevated calcemia; creatinine higher than 0.2 mM; elevated serum uric acid; eGFR < 60 mL/min;
[38]	Mosayebi et al., 2011	59	34 ± 9	Inclusion: MS; age 18–60 y; at least one relapse in the 12 months; more than three lesions on spinal or brain MRI; baseline EDSS score lower than 4.0. Exclusion: patients with CIS or progressive MS; clinical relapses during the study; use of digitalis; vitamin D supplementation; drug abuse; any condition pre-disposing to hypercalcemia; nephrolithiasis or renal impairment; pregnancy or lack of contraception; refusal to restrict dietary calcium.
[40,47]	Kimball et al., 2007 Burton et al., 2010	49	41 (range 22–54)	Inclusion: MS; age 18–55 y; EDSS score lower than 7.0. Exclusion: use of steroids within 30 days; relapse within 60 days; chemotherapy within 12 months; pregnancy/inadequate contraception; vitamin D intake higher than 4000 IU daily; serum 25(OH)D level > 150 nM; lymphoma or granulomatous disease; cardiac arrhythmia; kidney impairment; altered calcium metabolism.
[41]	Wingerchuk et al., 2005	15	36 (range 22–44)	Inclusion: RRMS; age 18–65 y; EDSS score 0–5.0; at least one clinical exacerbation in the previous year; contraindication to or patient’s desire against treatment with IFN-β and glatiramer acetate. Exclusion: progressive MS; use of glatiramer acetate, IFN-β, corticosteroid or immunosuppressive treatment within the previous two months; use of vitamin D supplementation or digitalis; any condition predisposing to hypercalcemia; nephrolithiasis or renal insufficiency; pregnancy or lack of contraception method; and refusal to restrict dietary calcium.

**Table 4 nutrients-15-01945-t004:** Intervention protocols and effects of the vitamin D supplementation on 25(OH)D blood levels.

Refs.	First Author, Year	Number of Included Participants	Intervention	Duration (Months)	Effect on 25(OH)D Blood Levels
[22]	Dörr et al., 2020	53	Oral vitamin D3 20,400 vs. 400 IU daily	18	From 48 (range 18–133) to 155 (130–200) nM
[21,43]	Hupperts et al., 2019 Rolf et al., 2017	229	Oral vitamin D3 6670 IU daily for 1 month, followed by 14,007 IU daily for 11 months	12	From 50 (IQR 35–75) to 220 (160–250) nM
[23]	Camu et al., 2019	129	Oral vitamin D3 100,000 IU every 2 weeks	24	From 49 ± 18 to 157 nM
[31]	Kotb et al., 2019	35	Oral vitamin D3 10,000 IU daily	12	From 23 ± 9.8 to 86 ± 7.3 nM
[32]	Darwish et al., 2017	88	Oral vitamin D3 10,000 IU daily	3	From 40 ± 16 to 123 ± 37 nM
[27]	O’Connell et al., 2017	67	Oral vitamin D3 5000 or 10,000 IU daily	6	From 53 to 129 or 168 nM
[28]	Laursen et al., 2016	134	Oral vitamin D3 2000 or 3000 or 4000 IU daily	6	From 34 to 33 (24–41) nM
[39]	Sotirchos et al., 2016	40	Oral vitamin D3 10,000 vs. 400 IU daily	6	From 68 ± 22 to 155 nM
[35,44]	Farsani et al., 2015	32	Oral vitamin D 50,000 IU weekly	2	From 31 ± 15 to 107 ± 43 nM
[36]	Etemadifar et al., 2015	15	Oral vitamin D3. 50,000 weekly from 12 to 16 weeks of gestation	5–6	From 38 ± 7 to 84 ± 38 nM
[33]	Achiron et al., 2015	158	Oral 1α(OH)D3 1 µg daily	6	Not provided
[34,45]	Golan et al., 2013a and 2013b	45	Oral vitamin D3. 75,000 IU every 3 weeks vs. 800 IU daily	12	From 48 ± 14 to 68 ± 11 or 123 ± 32 nM
[29,46]	Soilu-Hänninen et al., 2012; Hänninen et al., 2020	66	Oral vitamin D3 20,000 IU weekly	12	From 54 (range 19–82) to 110 (67–163) nM
[30]	Kampman et al., 2012	68	Oral vitamin D3 20,000 IU each week + 500 mg Ca if calcium intake < 500 mg daily	24	From 56 ± 29 to 121 nM
[37]	Shaygannejad et al., 2012	50	Oral 1,25(OH)_2_D3 0.50 µg daily	12	Not provided
[42]	Stein et al., 2011	23	Oral vitamin D2 7000 vs. 1000 IU daily	6	From 59 (IQR 47–61) to 120 (89–170)/ 69 (49–110) nM
[38]	Mosayebi et al., 2011	59	Intra-muscular vitamin D3 300,000 IU monthly	6	From 25 to 140 nM
[40,47]	Kimball et al., 2007 Burton et al., 2010	49	Oral vitamin D3 Escalating doses up to 40,000 IU daily for 7 months followed by 10,000 IU daily for 3 months, followed by 2 months without vitamin D + 1.2 g calcium throughout the study	12	From 73 (range 38–146) to 413 nM
[41]	Wingerchuk et al., 2005	15	Oral 1,25(OH)_2_D3. Escalating doses from 0.5 to 2.5 µg daily	11	Not provided

**Table 5 nutrients-15-01945-t005:** Summary of observations on relapses after vitamin D supplementation in MS patients.

Refs.	First Author, Year	Number Treated/Control Patients	Effects on Relapses
[22]	Dörr et al., 2020	28/25	No difference in the cumulative number of relapses between the high-dose (n = 5) and low-dose (n = 7) vitamin D3 groups (*p* = 0.6)
[21]	Hupperts et al., 2019	98/88	No difference in ARR between the vitamin D3-treated (0.28 ± 0.59) and placebo (0.41 ± 0.83) groups (*p* = 0.17). No difference in the proportion of patients who were free of relapses (78.8% vs. 75.0%, *p* = 0.47).
[23]	Camu et al., 2019	63/66 (post-hoc: 45/45)	Decreased rate ratio of relapses between the vitamin D3 and placebo groups (0.395, *p* = 0.01). A total of 19 relapses in the vitamin D3 group compared to 25 relapses in the placebo group. No difference in the time to first relapse [hazard ratio 0.801, 95% confidence interval (CI) 0.403–1.454, *p* = 0.43).
[27]	O’Connell et al., 2017	48/19	ARR of 0.2 (n = 1 relapse reported in the vitamin D3 group receiving 5000 UI/day). No statistical analysis in comparison with the placebo group.
[28]	Laursen et al., 2016	43/0	Each nM increase in 25(OH)D was independently associated with a −0.014 (95% CI −0.026 to −0.003) decrease in ARR (*p* = 0.02) in the vitamin D3-supplemented MS patients with baseline levels of 25(OH)D < 50 nM.
[39]	Sotirchos et al., 2016	19/21	One relapse in each treatment arm (high dose vs. low dose of vitamin D3) during the study. No comparative statistical analysis.
[36]	Etemadifar et al., 2015	6/9	Decrease in the mean number of relapses in the vitamin D3-treated and untreated groups, without statistical difference (−0.4, 95% CI −0.9–0.2, *p* < 0.06).
[33]	Achiron et al., 2015	80/78	A reduction in the number of relapses (10.5% vs. 33%, *p* = 0.006) and an increase in the proportion of relapse-free patients (90% vs. 67%, *p* = 0.007) in the 1α(OH)D3-treated group.
[34]	Golan et al., 2013	24/21	No significant change in ARR between the onset and the end of follow-up in the low-dose and high-dose vitamin D3-treated groups.
[29,46]	Soilu-Hänninen et al., 2012 Hänninen et al., 2020	34/32	Decrease in ARR in the vitamin D3-supplemented (−47%) and placebo (−45%) groups, without statistical difference between groups. No significant difference in the time to first relapse between groups (hazard ratio 1.12, 95% CI 0.41 to 3.1).
[30]	Kampman et al., 2012	35/36	No difference in the ARR between groups (ARR difference 0.06, 95% CI −0.08 to 0.20).
[37]	Shaygannejad et al., 2012	25/25	66% and 68% of patients remained relapse-free in the 1,25(OH)_2_D3 and placebo groups, respectively (odds-ratio 1.06, 95% CI 0.71–1.58). Significant decrease in ARR in the supplemented (−69%) and placebo (−62%) groups.
[42]	Stein et al., 2011	11/12	Relapses occurred in four (36.5%) patients in the high-dose vitamin D2 group vs. none in the low-dose vitamin D2 (*p* = 0.04).
[40,47]	Kimball et al., 2007 Burton et al., 2010	25/24	Decrease in ARR in the vitamin D3-supplemented (−41%) and control (−17%) groups, without statistical difference between groups (*p* = 0.09). No difference in the proportion of patients who were free of relapses (84% vs. 63%, *p* = 0.09).
[41]	Wingerchuk et al., 2005	15/0	Relapses occurred in four (27%) patients supplemented with 1,25(OH)_2_D3.

**Table 6 nutrients-15-01945-t006:** Summary of observations on the EDSS score after vitamin D supplementation in MS patients.

Refs.	First Author, Year	Number Treated/ Control Patients	Baseline EDSS Treated/ Control	Effects on the EDSS Score
[22]	Dörr et al., 2020	21/17	2.0 ± 3.5 (range) /2.0 (5.5)	No difference in the EDSS score at the end of follow-up between the high-dose and low-dose vitamin D3 groups [2.0 (range 3.5) vs. 2.0 (5.5), *p* = 0.26] and in the change from baseline [0 (4) vs. 0 (2.5)], *p* = 0.64.
[21]	Hupperts et al., 2019	98/88	Not provided	Similar proportion of patients who were free from EDSS progression at week 48 between the high-dose vitamin D3 and placebo groups (70.8% vs. 75.9%, *p* = 0.39).
[23]	Camu et al., 2019	63/66 (post hoc: 45/45)	1.7 ± 1.4/ 1.2 ± 1.2	Lower progression of the EDSS score in the vitamin D3 group (−0.06 ± 0.78) than in the placebo group (0.32 ± 0.87, 95% CI −0.614 to −0.043; *p* = 0.03). Mean decrease of EDSS of −0.003 per 1 nM increase of 25(OH)D concentration (95% CI −0.006 to −0.001; *p* = 0.006).
[31]	Kotb et al., 2019	35/0	2.2 ± 0.5	Higher mean EDSS at end of the follow-up (2.6 ± 0.5) compared to baseline (*p* = 0.02).
[27]	O’Connell et al., 2017	48/19	0.9 ± 1.0/ 0.9 ± 1.2/ 0.4 ± 0.5	No significant difference in the EDSS score between groups at any time point. No quantitative data provided.
[35]	Farsani et al., 2015	32/0	2.1 ± 1.1	Decrease in the mean EDSS score after vitamin D treatment to 1.89 ± 1.08 (*p* = 0.0036). No correlation between the EDSS score and 25(OH)D levels.
[36]	Etemadifar et al., 2015	6/9	1.2 ± 0.3/ 1.3 ± 0.4	Lower mean EDSS score in the vitamin D3-supplemented group compared to the routine care group at month 6 after delivery (1.1 ± 0.2 vs. 1.7 ± 0.6, *p* < 0.05).
[33]	Achiron et al., 2015	80/78	2.5 ± 1.6/ 2.8 ± 1.6	No significant change in the EDSS score in patients treated with 1,25(OH)_2_D3 or the placebo (0.06 vs. 0.31, *p* = 0.10).
[34,45]	Golan et al., 2013a et 2013b	15/15	2.9 ± 2.0/ 3.6 ± 2.2	No significant change in the EDSS score in each group. EDSS at the end of vitamin D3 treatment: 3.8 ± 2 vs. 3.4 ± 2.4 (no statistical analysis).
[29,46]	Soilu-Hänninen et al., 2012 Hänninen et al., 2020	32/30	2.0 ± 0.2/ 1.5 ± 1.2	Slight decrease in the mean EDSS score in the vitamin D3 group (to 1.8 ± 0.2). Similar EDSS score in the placebo group during the study (to 1.6 ± 1.3, *p* = 0.071).
[30]	Kampman et al., 2012	35/36	Not provided	No significant difference between groups in the absolute difference in EDSS (−0.01, 95% CI −0.35 to 0.35, *p* = 0.97).
[37]	Shaygannejad et al., 2012	25/25	1.6 ± 0.7/ 1.7 ± 1.2	No significant difference in the EDSS score at the end of treatment between the 1,25(OH)_2_D3 and placebo groups (1.63 ± 0.70 vs. 1.94 ± 1.41). Lower progression in the EDSS score during the study in the 1,25(OH)_2_D3 group than in the placebo group (difference 0.21, *p* < 0.05).
[42]	Stein et al., 2011	11/12	2.5 (IQR 2–4)/ 2.0 (1.0–3.0)	Higher EDSS score at the end of the follow-up in the high-dose vitamin D2 group: 3 (IQR 2–4) vs. 2 (1–2) in the low-dose group (*p* = 0.04).
[38]	Mosayebi et al., 2011	26/33	2.1 ± 1.2/ 2.5 ± 1.1	No significant difference in the EDSS score between the vitamin D3-treated and control groups (EDSS scores after treatment: 2.31 ± 1.3 and 2.67 ± 1.25, respectively).
[40,47]	Kimball et al., 2007 Burton et al., 2010	25/24	1.5 ± 1.6/ 1.2 ± 1.6	No significant difference in the change in the EDSS score during the trial between the vitamin D3 and placebo groups (−0.23 vs. 0.30, *p* > 0.05). Lower proportion of patients who completed the trial with a higher EDSS score in the vitamin D3 group than in the placebo group (8 vs. 38%, *p* = 0.02).
[41]	Wingerchuk et al., 2005	15/0	1.9 (range 0–4.0)	Mean EDSS at the end of the study: 2.2. Median EDSS change = 0 (range −1.0 to 2.0). 27% of patients worsened ≥1 EDSS point.

**Table 7 nutrients-15-01945-t007:** Summary of observations on the MRI lesions in MS patients after vitamin D supplementation.

Refs.	First Author, Year	No Treated/Control Patients	Effects on MRI Lesions
[22]	Dörr et al., 2020 EVIDIMS trial	21/17	Lower number of T2-weighted lesions at month 18 in the high-dose vitamin D3 group compared to the low-dose vitamin D3 group [53.4 ± 7.3 (SE) vs. 84.1 ± 13.5], but the change from baseline did not differ between the two groups (1.3 ± 0.1 vs. 2.1 ± 1.4, *p* = 0.15). The change from baseline in the T2 lesion volume did not differ between the two groups (0.1 ± 0.1 vs. 0.2 ± 0.3 mL, *p* = 0.98). The cumulative number of new Gd lesions was 2 and 14 in the high-dose and low-dose groups (*p* = 0.09).
[21]	Hupperts et al., 2019 SOLAR trial	98/88	32% reduction in the number of new Gd-enhancing or new/enlarging T2 lesions in the vitamin D3 group compared to the placebo at week 48 (incidence rate ratio 0.68, *p* = 0.005). No difference between treatments in the proportion of patients free from new T1 hypointense lesions at week 48 (79 vs. 64%, *p* = 0.30). Higher difference in the change in the total volume of T2 lesions (*p* = 0.035).
[23]	Camu et al., 2019 CHOLINE trial	44/41	Significant reduction in the volume of hypointense T1-weighted lesions (−312 mm^3^, 95% CI −596 to −29, *p* = 0.03) and new T1 lesions (rate ratio 0.494, 95% CI 0.267–0.913, *p* = 0.03) in vitamin D3-treated patients compared to the placebo. No difference in the number of Gd-enhancing T1 and new T2 lesions.
[27]	O’Connell et al., 2017	19/7	No difference in the number of new T2 lesions or new Gd-enhancing lesions between groups at month 6. The number of CIS patients with new disease activity based on MRI was 50% (10,000 IU/day of vitamin D3), 56% (5000 IU/day of vitamin D3), and 43% (placebo).
[29]	Soilu-Hänninen et al., 2012 Finnish Vitamin D Study	32/30	Lower number of Gd-enhancing T1 lesions in the vitamin D3-treated group compared to the placebo at month 12 (0.1 ± 0.2 vs. 0.7 ± 3.5, *p* = 0.004). No difference in the number of new/enlarging T2 lesions (0.5 ± 1.0 vs. 1.1 ± 2.2, *p* = 0.29) and in the change in the total volume of T2 lesions (83 ± 128 vs. 287 ± 283 mm^3^, *p* = 0.105).
[42]	Stein et al., 2011	11/12	No difference in the cumulative number of Gd-enhancing lesions in the high-dose vitamin D2-treated group compared to the low-dose vitamin D2 group (14 ± 0.2 vs. 0.7 ± 3.5, *p* = 0.004), in the number of patients with new lesions and in the change in the total volume of T2 lesions [−330 (IQR −950 to −30) vs. −95 (−310 to −25) mm^3^, *p* = 0.6).
[38]	Mosayebi et al., 2011	26/33	No difference in the number of Gd-enhancing lesions between the MS patients taking 300,000 IU monthly of vitamin D3 or the placebo (from 1.5 ± 1 to 1.9 ± 0.7 and 1.6 ± 0.8 to 2.0 ± 1.0, respectively).
[47]	Kimball et al., 2007	12/0	Reduction in the mean number of Gd-enhancing lesions after vitamin D3 supplementation (from 1.75 ± 1.42 to 0.83 ± 0.72, *p* = 0.03).
[41]	Wingerchuk et al., 2005	15/0	New T2-weighted lesions were found in 43% of patients at week 24 and in 29% at week 48.

## Data Availability

The data presented in this study are available on request from the corresponding author.

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
