# Peer review of "Clinical and Imaging Outcomes after Vitamin D Supplementation in Patients with Multiple Sclerosis: A Systematic Review"

_nutrients, 2023, doi:10.3390/nu15081945_

Round 1
Reviewer 1 Report
This is an interesting and well-written systematic review. I suggest the publication of this manuscript after address the following issue:
-Please add some examples and discussion of images of magnetic resonance imaging (MRI) related to patients with MS and vitamin D supplementation. These images will help the readers.
Author Response
“This is an interesting and well-written systematic review. I suggest the publication of this manuscript after address the following issue:
-Please add some examples and discussion of images of magnetic resonance imaging (MRI) related to patients with MS and vitamin D supplementation. These images will help the readers.”
We would like to thank the Reviewer for his positive feedback on our manuscript. We fully agree with the Reviewer that we can add representative MRI images in order to help the readers. We therefore added some representative MRI images in the Figure 1 to illustrate typical hyperintense lesions in brain of patients with MS. We also improved the description of MRI lesions in MS patients by adding some sentences in the introduction section of the revised manuscript. All revisions made to the manuscript are marked up using the “Track Changes” function, as required by the Editor.
Reviewer 2 Report
Major comments:
1. This systematic review came to the overall conclusion that vitamin D3 supplementation had no statistically significant effect on MS relapse. This result may be true but needs more critical inspection. For example, if the study participants were already sufficiently supplemented, i.e., not vitamin D deficient, and additional vitamin D3 supplementation will not result in any significant effect.
2. Please discuss comparable vitamin D intervention study with the focus on other diseases, such VITAL.
Minor comments:
1. When you use the term "vitamin D" please clarify, if you mean "vitamin D3", 25(OH)D3" or "1,25(OH)2D3".
2. Line 54: There is only one VDR, please do not use plural.
3. Please be consistent with vitamin D metabolite nomenclature. If you start with "vitamin D3", then please used 25(OH)D3" and "1,25(OH)2D3" (not calcitriol).
4. Please define all abbreviations at first time use and use them then consistently, i.e. please no double definition.
5. Please be consistent to use either "nmol/l" or "nM"
6. "20,400 IU", is this a typo?
Author Response
We would like to thank the Reviewer for his insightful comments, which have helped us to improve the manuscript. All revisions made to the manuscript are marked up using the “Track Changes” function, as required by the Editor.
“Major comments:
- This systematic review came to the overall conclusion that vitamin D3 supplementation had no statistically significant effect on MS relapse. This result may be true but needs more critical inspection. For example, if the study participants were already sufficiently supplemented, i.e., not vitamin D deficient, and additional vitamin D3 supplementation will not result in any significant effect.
We thank the Reviewer for having stressed this concern. We fully agree that it is relevant in the field of vitamin D supplementation to know the vitamin D status at baseline. In fact, vitamin D supplementation may be more effective at preventing relapses if MS participants are vitamin D deficient at baseline. We had reported in the Table 4 the 25(OH)D blood levels at baseline and after supplementation for the selected studies. In the discussion section, we had mentioned that the MS participants in CHOLINE RCT (showing a positive effect of vitamin D3 on relapses) and in SOLAR RCT (showing a lack of efficiency of vitamin D3) had similar progression of 25(OH)D levels from baseline to the end of supplementation. However, following the Reviewer’s advices, we have reinforced the discussion about vitamin D status in MS participants at baseline in the discussion section.
- Please discuss comparable vitamin D intervention study with the focus on other diseases, such VITAL.
We thank the Reviewer for having proposed this suggestion. We fully agree that we can discuss the interest of vitamin D supplementation on other diseases, especially on other autoimmune diseases than MS. We acknowledge the Reviewer for having suggested the interesting results yielded by the VITAL RCT. In this large RCT, Hahn et al. (BMJ 2022;376:e066452) concluded that vitamin D supplementation for five years reduced the incidence of autoimmune diseases by 22%. We also mentioned that there is some evidence that vitamin D supplementation during early infancy may reduce the incidence of type 1 diabetes, another autoimmune disease. These results reinforce the link between vitamin D and autoimmune diseases, and therefore add to the rationale of studying the effect of vitamin D in MS patients. We therefore added this relevant background at the end of the introduction section.
Minor comments:
- When you use the term "vitamin D" please clarify, if you mean "vitamin D3", 25(OH)D3" or "1,25(OH)2D3".
We have clarified the form of vitamin D throughout the revised manuscript (in addition to the Table 4).
- Line 54: There is only one VDR, please do not use plural.
We have corrected this point in the revised manuscript.
- Please be consistent with vitamin D metabolite nomenclature. If you start with "vitamin D3", then please used 25(OH)D3" and "1,25(OH)2D3" (not calcitriol).
We have therefore used the following nomenclature throughout the revised manuscript: vitamin D2 or vitamin D3 for the native forms of vitamin D, 25(OH)D for 25-hydroxyvitamin D, 1α(OH)D3 for alfacalcidiol, and 1,25(OH)2D3 for calcitriol.
- Please define all abbreviations at first time use and use them then consistently, i.e. please no double definition.
We carefully checked all abbreviations, and added some corrections throughout the revised manuscript.
- Please be consistent to use either "nmol/l" or "nM"
We have carefully checked the expression of units, and used “nM” throughout the revised manuscript.
- "20,400 IU", is this a typo?”
We understand that “20,400 IU” may appear as a typo at first reading. In fact, the vitamin D3 dose used in the EVIDIMS RCT was actually 20,400 IU/day in the high-dose group (and 400 IU/day in the low-dose group). MS patients in the high-dose group took each day 1 mL of vitamin D3 (corresponding to 20,000 IU) plus 1 tablet of 400 IU, whereas MS patients in the low-dose group took 400 IU daily (1 tablet of 400 IU + 1 mL placebo oil).
Round 2
Reviewer 2 Report
The manuscript improved sufficiently